🔓 | **Open Peer Review** | Mycology | Research Article

# Loss of the *Aspergillus fumigatus* spindle assembly checkpoint components, SldA or SldB, generates triazole heteroresistant conidial populations

Ashley V. Nywening,[1,2] Harrison I. Thorn,[1,3] Jinhong Xie,[1,3] Adela Martin-Vicente,[1] Xabier Guruceaga,[1] Wenbo Ge,[4] John G. Gibbons,[5] Jarrod R. Fortwendel[1]

**ABSTRACT** The opportunistic pathogen *Aspergillus fumigatus* is the chief causative agent of human invasive filamentous fungal infections. Triazoles, the primary therapeutic options to combat invasive aspergillosis (IA), target the biosynthesis of ergosterol, a vital component of the fungal cell membrane. Unfortunately, resistance to this class of medical therapeutics has arisen globally and now threatens the future usefulness of these compounds for antifungal treatment. Infection with *A. fumigatus* with acquired triazole resistance increases an already high associated mortality rate and reduces the limited arsenal of therapeutic options to combat IA. How this specific fungal pathogen obtains resistance remains poorly understood. In this study, we show that loss of the previously uncharacterized *A. fumigatus* spindle assembly checkpoint (SAC) components, SldA or SldB, resulted in a heteroresistance phenotype to multiple mold active medical triazoles and to compounds inhibiting ergosterol biosynthesis at points upstream of the triazole target, Cyp51A. Consistent with conserved roles in mitotic fidelity, loss of either component resulted in the production of conidia characterized by an increased genome size, suggestive of potential aneuploidy development. Interestingly, we find that heteroresistance of the Δ*sldA* or Δ*sldB* conidial populations was only evident in response to ergosterol biosynthesis pathway inhibition and not seen with other external stress. Our findings support the hypothesis that specific links exist between SAC function and resistance to ergosterol biosynthesis perturbation in *A. fumigatus*.

**IMPORTANCE** The rising threat of antifungal resistance in *Aspergillus fumigatus*, a filamentous fungal species which remains one of the leading causes of human invasive infections, is an increasingly relevant concern to public health worldwide. The mode and mechanism of triazole resistance acquisition remain an understudied issue for this opportunistic pathogen. This work uncovers a novel role for a functional spindle assembly checkpoint in maintaining susceptibility to ergosterol biosynthesis inhibitors, including the triazole antifungal drug class.

**KEYWORDS** *Aspergillus fumigatus*, triazole resistance, antifungal susceptibility, antifungal tolerance, ergosterol biosynthesis, spindle assembly checkpoint, aneuploidy

Aspergillus fumigatus is the leading cause globally of invasive filamentous fungal infections within susceptible human populations (1, 2). A saprobic organism, *A. fumigatus*, grows on a variety of substrates (3, 4). The species produces asexual reproductive structures called conidia in abundance, which are designed primarily to be easily spread by air currents (5). The average individual will inhale about 300 conidia every day from this ubiquitous species (4, 6, 7). Immunocompetent hosts can clear inhaled conidia through phagocytosis by resident innate immune cells (5). However,

**Peer Reviewers** Gustavo H. Goldman, Universidade de Sao Paulo, Sao Paulo, Brazil; W. Scott Moye-Rowley, The University of Iowa, Iowa City, Iowa, USA

Address correspondence to Jarrod R. Fortwendel, jfortwen@uthsc.edu.

The authors declare no conflict of interest.

See the funding table on p. 17.

conidia which are not cleared may germinate and initiate an invasive infection. Invasive aspergillosis (IA) represents the most severe form of *Aspergillus* infection, producing a high rate of mortality ranging from 30% to approaching 90% for some patient populations (5, 7–9). Cases of IA have increased worldwide in recent history likely due to increases in susceptible populations, such as patients with neutropenia and hematological malignancies, transplant recipients, those on steroids or other immune-suppressing medications, and those with underlying primary disease states such as COPD, cystic fibrosis, tuberculosis, sarcoidosis, or tissue damage (2, 5, 10). Cases of IA also occur in patients with viral respiratory tract infections, including influenza and coronavirus disease 2019, and can occur even in healthy individuals if exposed to a sufficiently high inoculum of fungal spores (2, 10–12). Aspergilloses are associated with approximately 14,000 hospitalizations and an annual healthcare burden of between $600 million and $1.2 billion in the United States alone (9, 13).

Of the limited number of approved therapeutics, the triazoles remain the primary option recommended to combat *Aspergillus* infections, with voriconazole representing the primary drug of choice for invasive disease (14). The triazole antifungals are fungicidal against *A. fumigatus*, can also be administered orally rather than only intravenously, and can be given for months with lower treatment-associated adverse events (14–16). In fact, extended triazole therapy, either for prophylaxis or in response to chronic infections, occurs often (15, 17). Unfortunately, extended periods of patient treatment with triazole antifungals consequently increase the risk of the development of triazole resistance (18, 19). The cost of IA to public health is increased by the involvement of a triazole-resistant isolate. Whether acquired from an environmental source or developed within a patient's own system as a consequence of therapy, infection with *A. fumigatus*, which has adapted to be resistant to triazole therapy, increases the expected risk of patient mortality (20, 21). Since the first recorded incidence of resistant infection in 1997, encounters with triazole-resistant *A. fumigatus* have increased (22, 23). Recently, *A. fumigatus* was one of only four species placed in the critical category, the highest priority tier of the list, alongside *Cryptococcus neoformans*, *Candida auris*, and *Candida albicans* on the World Health Organization's fungal priority pathogens list, a report on the 19 fungi which represent the greatest threats to public health (24, 25).

Triazole resistance in *A. fumigatus* remains poorly understood. Of the mechanisms which are currently known or suspected to contribute to triazole resistance in *A. fumigatus*, the majority involve alterations to the expression of gene transcript for the enzymes targeted by these antifungals, *cyp51A* and *cyp51B*, or a mutation which affects the ability of the compounds to associate with the Cyp51A enzyme (2, 26). Such mutations reduce the effectiveness of the drug. Alterations which impact Cyp51A are predominantly associated with resistance, whereas the influence of transcriptional or sequential Cyp51B changes is less certain (16, 26). Additional mechanisms which are implicated in *A. fumigatus* azole resistance include overexpression of drug efflux pumps that actively remove the drug from fungal cells, as well as mutations in the HMG-CoA reductase ortholog, Hmg1, or in the CCAAT-binding transcription factor complex subunit, HapE (27–30). However, there are many resistant *A. fumigatus* isolates for which no known resistance mechanism is present (31, 32).

Protein kinases are known to play diverse roles in eukaryotic cellular processes, including responses to antifungal stress, via reversible phosphorylation events (33–36). Here, we screened a library of *A. fumigatus* protein kinase disruption mutants constructed by our lab in a previous study to identify novel signaling events important for response to triazole stress (37). Out of 118 mutant strains, only disruption of two genes predicted to encode *A. fumigatus* protein kinases altered voriconazole susceptibility by at least fourfold. One of these predicted kinase-encoding genes encodes an ortholog of the spindle assembly checkpoint (SAC) kinase, SldA, of the model organism *Aspergillus nidulans*. Our findings show that deletion of *sldA* or the putative SldA mitotic checkpoint complex (MCC) binding partner, *sldB*, results in the production of heteroresistant conidial progeny exhibiting decreased susceptibility to compounds that inhibit

ergosterol biosynthesis. Loss of these SAC components also resulted in the production of conidia with increased genome size. Surprisingly, the ability of Δ*sldA* and Δ*sldB* strains to resist external stress is specific to ergosterol pathway inhibitors, as we find these strains to be susceptible to cell wall, oxidative, osmotic, and host stresses. As the SAC is a highly conserved cell cycle checkpoint, our findings support the hypothesis that specific links exist between maintenance of mitotic fidelity and resistance to ergosterol biosynthesis perturbation in *A. fumigatus*.

## RESULTS

### Identification of protein kinases regulating triazole susceptibility in *A. fumigatus*

To identify molecular mechanisms underpinning triazole stress responses, we screened an *A. fumigatus* protein kinase gene disruption library for increased or decreased voriconazole susceptibility (37). All 118 protein kinase gene disruption mutants were analyzed by Clinical and Laboratory Standards Institute M38-A2 broth microdilution minimum inhibitory concentration (MIC) assay (37), revealing only two mutants with voriconazole MIC changed by at least fourfold. Surprisingly, both of these mutant strains exhibited a higher MIC than the parental strain, indicating a reduction in susceptibility to the triazole (Fig. 1A and B). No mutant displayed hypersusceptibility. One of the identified mutants possessed a disruptive mutation in a previously characterized gene encoding an *A. fumigatus* cyclin-dependent kinase, Ssn3. Loss of *ssn3* was previously shown to decrease triazole susceptibility in *A. fumigatus* (38), validating our MIC assay results. The second disruption mutant identified in our assays possessed a disruption in a previously uncharacterized *A. fumigatus* gene predicted to encode the singular ortholog of the SAC kinase, SldA. The predicted *A. fumigatus* SldA protein shares significant sequence similarity to the SAC kinase orthologs from other fungal species, including *A. nidulans* SldA (67.28% identity and 99% coverage), *C. albicans* BUB1 (30.54% identity and 72% coverage), and the two orthologous proteins in *Saccharomyces cerevisiae*, BUB1 (25.28% identity and 92% coverage) and Mad3p (27.58% identity and 32% coverage) (39, 40).

### The SAC kinase is not required for normal growth and development in *A. fumigatus*

To determine the impacts of *sldA* mutation on growth and development and to ensure that any phenotypes resulting from *sldA* gene disruption were truly the result of a loss of function, we next generated a complete *sldA* gene deletion mutant via CRISPR/Cas9 gene editing (Fig. S1A) (41). Simultaneously, we generated a deletion of the gene encoding the putative SldA-binding protein, SldB (Fig. S1C) (39). The putative *A. fumigatus* SldB (encoded by the gene AFUB_076660 in the strain A1163) is a predicted ortholog of *A. nidulans* SldB, which is known to bind to the single *A. nidulans* SldA kinase to regulate its function in the MCC (Fig. 1C) (42). *A. fumigatus* SldB is also a predicted ortholog of the *S. cerevisiae* BUB3 (encoded by the gene YOR026W in the reference strain S288C) (39). In *S. cerevisiae*, which like many eukaryotes expresses two paralogous SAC kinases known as BUB1 (YGR188C in S288C) and MAD3 (YJL013C in S288C) (39), Bub3p is known to bind to the kinase Mad3p to regulate its function in the MCC complex, and from *S. cerevisiae* to humans, BUB3 is known to complex with BUB1 to aid kinetochore interactions in response to checkpoint activation (39, 43). Therefore, loss of the *A. fumigatus* SldB would be expected to impact the SAC similarly to loss of the SldA kinase. Both deletion mutants were then complemented by reinsertion of the entire *sldA* or *sldB* ORF into each native locus (Fig. S1B and D).

Growth assays on minimal media revealed no gross differences in colony morphology for the *sldA* (Δ*sldA*) or *sldB* (Δ*sldB*) deletion strains when compared to the wild-type parental strain, CEA10 (Fig. 2A). Quantitation of colony diameter during the initial 96 h of culture revealed statistically significant, yet minor, reductions in vegetative growth when

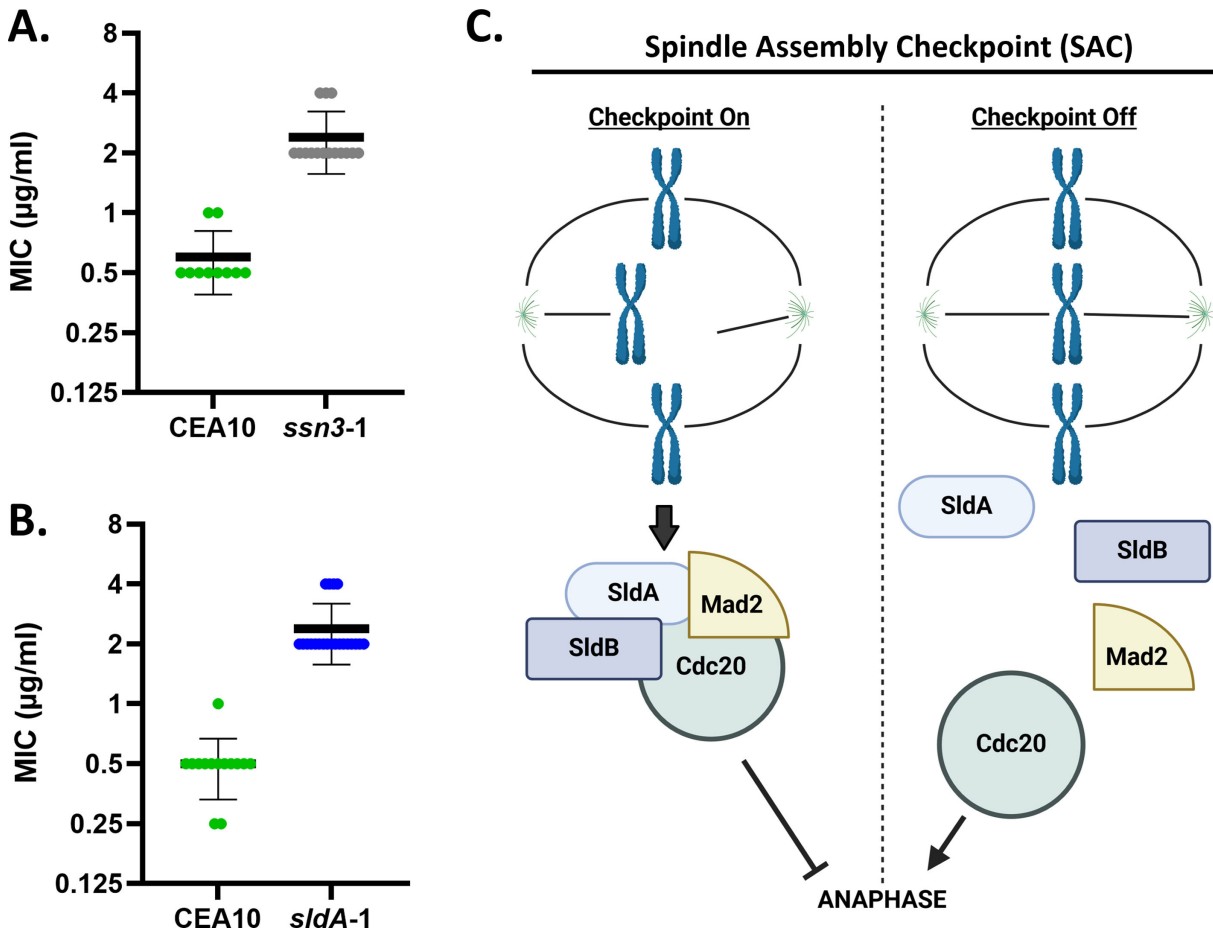

**FIG 1** Disruption of the *sldA* and *ssn3* genes results in reduced triazole susceptibility. (**A**) MIC assay comparing the wild type *A. fumigatus* progenitor strain (CEA10/A1163) voriconazole susceptibility to that of the *ssn3* gene disruption strain (*ssn3-1*). (**B**) MIC assay comparing CEA10 progenitor susceptibility to that of the *sldA* gene disruption strain (*sldA-1*). The MIC was defined as the concentration of triazole that inhibited 100% of growth. MICs were determined in Roswell Park Memorial Institute 1640 after 48 h at 37°C, according to the Clinical and Laboratory Standards Institute M38 document. Individual results from multiple experiments are shown within each graph. Bold central line denotes the mean MIC. Error bars represent standard deviations. (**C**) Schematic of SldA and SldB involvement in the SAC.

comparing mutant and wild-type strains (Fig. 2B). When growth was measured as biomass accumulation in submerged culture, no significant reduction in biomass was noted between mutant and wild-type strains (Fig. 2C). An assay to compare the rate of conidial germination revealed that the strains exhibited similar germination kinetics and reached a similar percent germination after 16 h of culture (Fig. 2D). Therefore, SldA and SldB are not required for maintenance of normal growth and development in *A. fumigatus*.

## Loss of the *sldA* or *sldB* gene results in increased susceptibility to benomyl but does not alter nuclear distribution

Related to its functional role in the SAC, deletion of the *A. nidulans sldA* gene results in hypersusceptibility to the benzimidazole spindle poison, benomyl (44). To determine if the functions of SldA and SldB for the SAC pathway are likely conserved in *A. fumigatus*, susceptibility to benomyl was determined for both Δ*sldA* and Δ*sldB*. On solid agar, both mutant strains exhibited an increase in susceptibility to benomyl, with the Δ*sldA* mutant displaying a more severe phenotype than Δ*sldB* (Fig. 3A). Examination of benomyl susceptibility in submerged culture by broth microdilution assay resulted in a similar outcome for both strains (Fig. 3B). These results suggest conserved functions for SldA

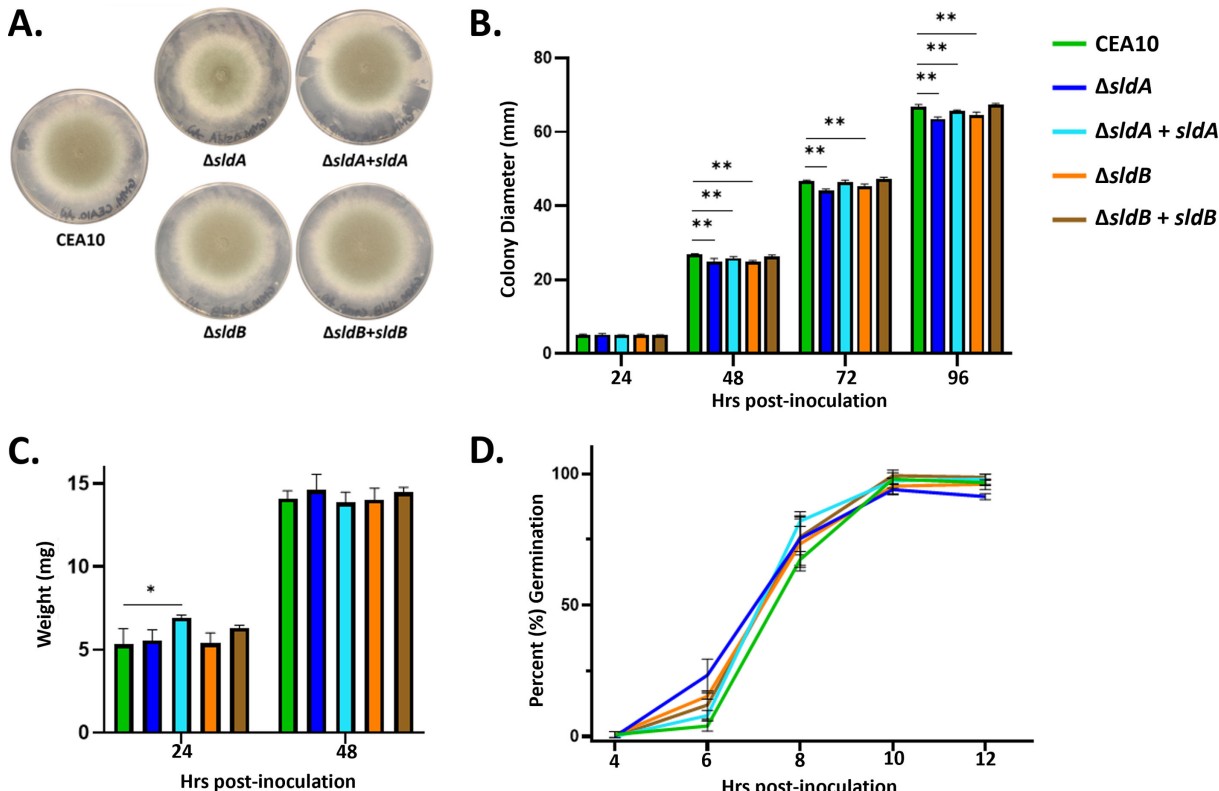

FIG 2 The SAC pathway components, SldA and SldB, are not required for normal growth and development in *A. fumigatus*. (**A**) Colony morphologies of CEA10, Δ*sldA*, Δ*sldA* complement (Δ*sldA* + *sldA*), Δ*sldB*, and Δ*sldB* complement (Δ*sldB* + *sldB*) strains cultured for 96 h on glucose minimal medium (GMM). (**B**) Quantitation of colony diameter on solid GMM over time. (**C**) Quantitation of biomass accumulation in liquid GMM at 24 and 48 h timepoints. (**D**) Quantitation of germination assay in liquid GMM. A minimum of three independent tests per strain were performed for each experiment. For panels B–D, statistical analysis was performed using one-way analysis of variance with Tukey's test for multiple comparisons. Error bars represent standard deviations. *P* values are considered significant at *$P < 0.05$, **$P < 0.01$.

and SldB in *A. fumigatus* and that microtubule stress in combination with loss of the SldA kinase is more poorly tolerated than in combination with loss of the SldB protein. To assess if the roles for SldA or SldB in resistance to microtubule stress are associated with basal defects in the progression of mitosis, we next quantified the average number of nuclei present within interseptal hyphal compartments. Observation of the distribution of nuclei throughout hyphae using propidium iodide (PI) staining did not reveal gross misdistribution (Fig. 4A). Quantitative assessment of the average number of nuclei per

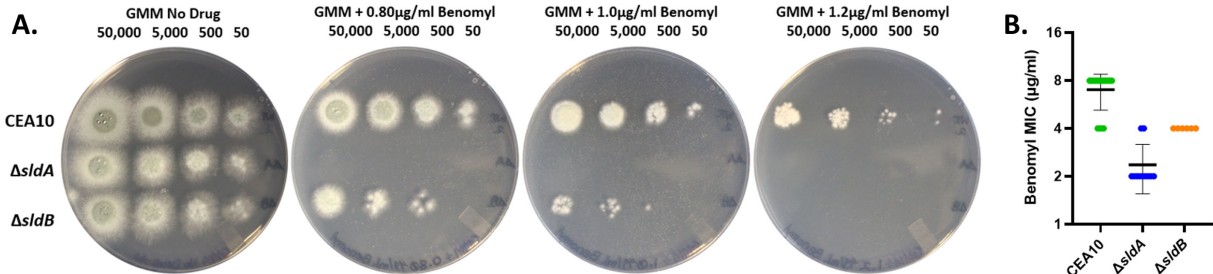

FIG 3 Loss of *sldA* or *sldB* results in increased susceptibility to benomyl. (**A**) Spot-dilution assay plates comparing growth of the CEA10 progenitor strain to that of the *sldA* gene deletion strain and the *sldB* gene deletion strain on GMM in the absence (GMM no drug) and presence of benomyl at indicated concentrations. All images were acquired after 48 h incubation at 37°C. (**B**) MIC assay of the CEA10, Δ*sldA*, and Δ*sldB* strains by broth microdilution in GMM broth. MICs were determined after 48 h at 37°C. Individual results from multiple experiments are shown within the graph. The bold central line denotes the mean MIC. Error bars represent standard deviations.

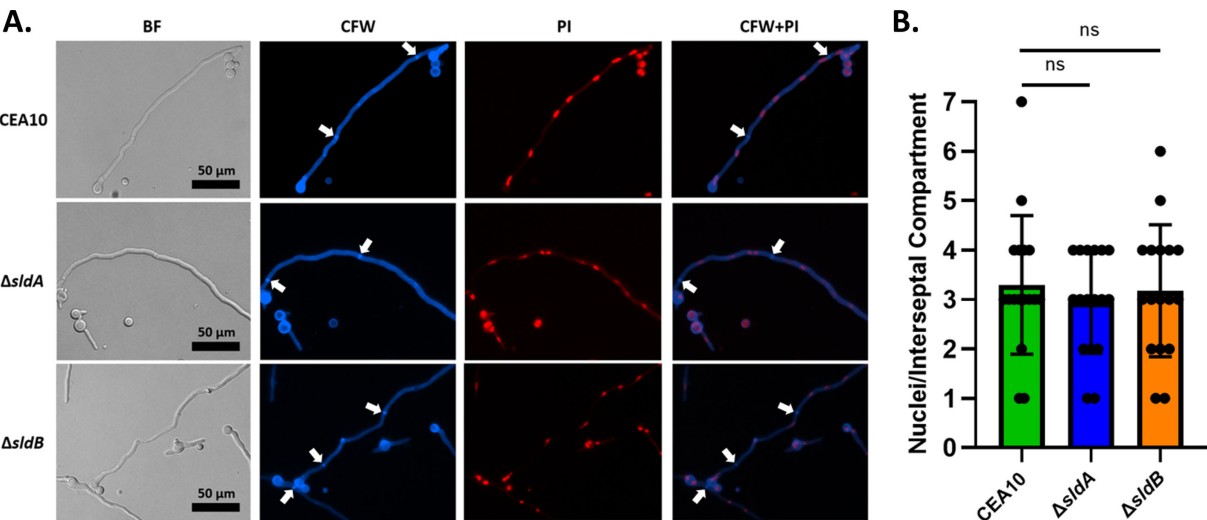

**FIG 4** The SAC pathway components, SldA and SldB, are not required for normal nuclear number or positioning in *A. fumigatus*. (**A**) Representative images of 16 h germlings stained with propidium iodide (PI) and Calcofluor White (CFW). White arrows indicate the location of septa. (**B**) Quantification of nuclear number per interseptal hyphal compartment (*n* = 50/strain). Error bars represent standard deviations. Individual results from multiple analyses of stained samples are shown within the graph. Statistical analysis by one-way analysis of variance with Dunnett's post hoc test.

interseptal compartment revealed no significant differences between the Δ*sldA* and Δ*sldB* strains and wild type (Fig. 4B). Therefore, *sldA* and *sldB* do not appear essential for normal nuclear distribution throughout hyphae.

## Loss of *sldA* or *sldB* specifically impacts susceptibility to inhibitors of ergosterol biosynthesis

As the *sldA* disruption mutant displayed decreased susceptibility to voriconazole, we next sought to define the impact of *sldA* or *sldB* deletion on susceptibility to a panel of ergosterol biosynthesis inhibitors (EBIs). The EBI antifungal compounds were chosen from multiple drug classes, including the triazoles (voriconazole, itraconazole, posaconazole, and isavuconazole), the allylamines (terbinafine) and the statins (fluvastatin) (Fig. 5A). Broth microdilution MIC assays revealed that the reduced voriconazole susceptibility phenotype originally detected in *sldA*-1 was also evident in Δ*sldA* (Fig. 5D). These data suggest that our original *sldA* disruption was, as expected, a loss-of-function mutation. Although the average voriconazole MIC increase upon loss of *A. fumigatus sldA* was only fourfold, we also found voriconazole susceptibility of an *Aspergillus nidulans sldA* deletion strain to be reduced, supporting this phenotype as a result of SAC kinase deletion in multiple *Aspergillus* spp. (Fig. S2). Furthermore, MIC assays revealed that both Δ*sldA* and Δ*sldB* mutant strains exhibited a consistent pattern of reduced susceptibility to each of the triazole compounds utilized (Fig. 5D through G). Interestingly, both Δ*sldA* and Δ*sldB* strains were also less susceptible than wild type to fluvastatin and terbinafine, indicating that loss of *sldA* or *sldB* impacts susceptibility to inhibitors of ergosterol biosynthesis in general (Fig. 5B and C). However, neither mutant exhibited an altered susceptibility to the polyene compound, amphotericin B, which, rather than inhibiting ergosterol biosynthesis, targets existing ergosterol within the fungal cell membrane (Fig. 5H). As loss of *sldA* or *sldB* generated resistance to EBIs in general, we predicted that the mechanisms underlying reduced triazole susceptibility in our mutants were likely not due to simple transcriptional upregulation of previously characterized triazole resistance genes such as the triazole target genes, *cyp51A* and *cyp51B*, or drug efflux pumps (2, 26). In agreement with this, analysis for differential gene expression of *cyp51A* and *cyp51B* and of the drug efflux pumps, *atrF* and *abcC/cdr1B*, revealed no major differences in gene expression, either at baseline or under sub-MIC voriconazole exposure, between Δ*sldA*

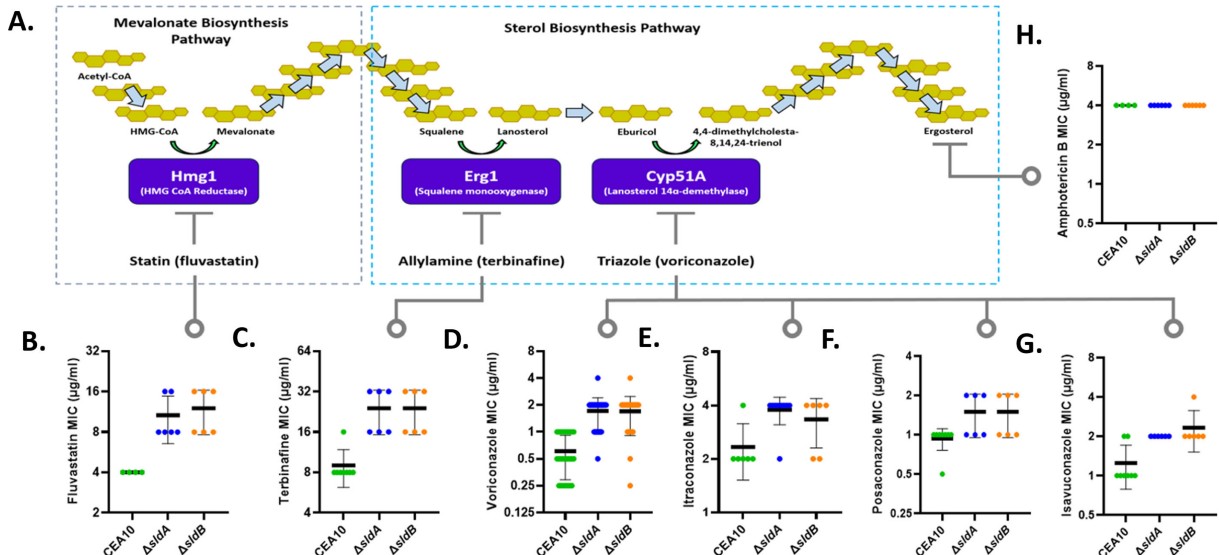

**FIG 5** Loss of gene *sldA* or *sldB* reduces susceptibility to inhibitors of ergosterol biosynthesis. (**A**) A schematic of the ergosterol biosynthesis pathway with specified points of chemical inhibition noted. MIC results are shown for the ergosterol biosynthesis inhibitors (**B**) fluvastatin, (**C**) terbinafine, (**D**) voriconazole, (**E**) itraconazole, (**F**) posaconazole, and (**G**) isavuconazole and for the ergosterol-binding polyene compound, (**H**) amphotericin B. Assays were performed as in Fig. 3B. Individual results from multiple experiments are shown within the graph. Bold central line denotes the mean MIC. Error bars represent standard deviations.

and CEA10 (Fig. S3). These findings suggested a potentially uncharacterized mechanism underlying triazole susceptibility loss when *sldA* is deleted.

To determine if the Δ*sldA* or Δ*sldB* mutants also displayed altered susceptibility to non-ergosterol biosynthesis pathway-related stress, MIC/minimum effective concentration (MEC) assays were repeated under multiple stress conditions. These conditions included the addition of the cell wall-targeting antifungal compound, caspofungin; the DNA damage inducer, MMS; the oxidative stress inducers, 4-nitroquinalone and paraquat; and the osmotic stressors NaCl and sucrose. Interestingly, none of the chosen stressors revealed a pattern of altered susceptibility in either the Δ*sldA* or Δ*sldB* mutant when compared to wild type (Fig. 6A through F). Employing the host niche as a stress-inducing environment, we also found that the Δ*sldA* and Δ*sldB* strains did not display altered pathogenic fitness in a murine model of invasive aspergillosis (Fig. 6G). Therefore, loss of *sldA* or *sldB* was not associated with a general alteration of susceptibility to any non-ergosterol biosynthesis stress applied.

## SAC component deletion generates conidia with higher average DNA content

Aneuploidy has been shown to provide either phenotypic resistance to triazoles or a heteroresistance phenotype wherein only a subset of fungal cells exhibits reduced triazole susceptibility (45–49). Analysis of micrographs taken from the wells of voriconazole MIC assays revealed that both Δ*sldA* and Δ*sldB* mutants exhibited a heteroresistance-like phenotype, with only a small subpopulation of conidia that form microcolonies at concentrations of triazole which are completely inhibitory to the wild type (Fig. 7). Flow cytometry to quantify DNA content is an established method utilized to assess genome size within populations of fungal cells or dormant fungal spores (50–52). Therefore, we performed flow cytometric analyses of fixed and stained conidia, as previously described (50). To validate the conidial DNA staining protocol, we first analyzed a wild-type population of triazole-susceptible wild-type conidia alongside stable diploid and haploid strains of the model yeast *S. cerevisiae* (Fig. 8A). Flow analysis produced the expected fluorescence signal for the control *S. cerevisiae* strains American Type Culture Collection 204508 and 201390, indicative of a haploid and a diploid genome, respectively, and which correspond to previously published results for these

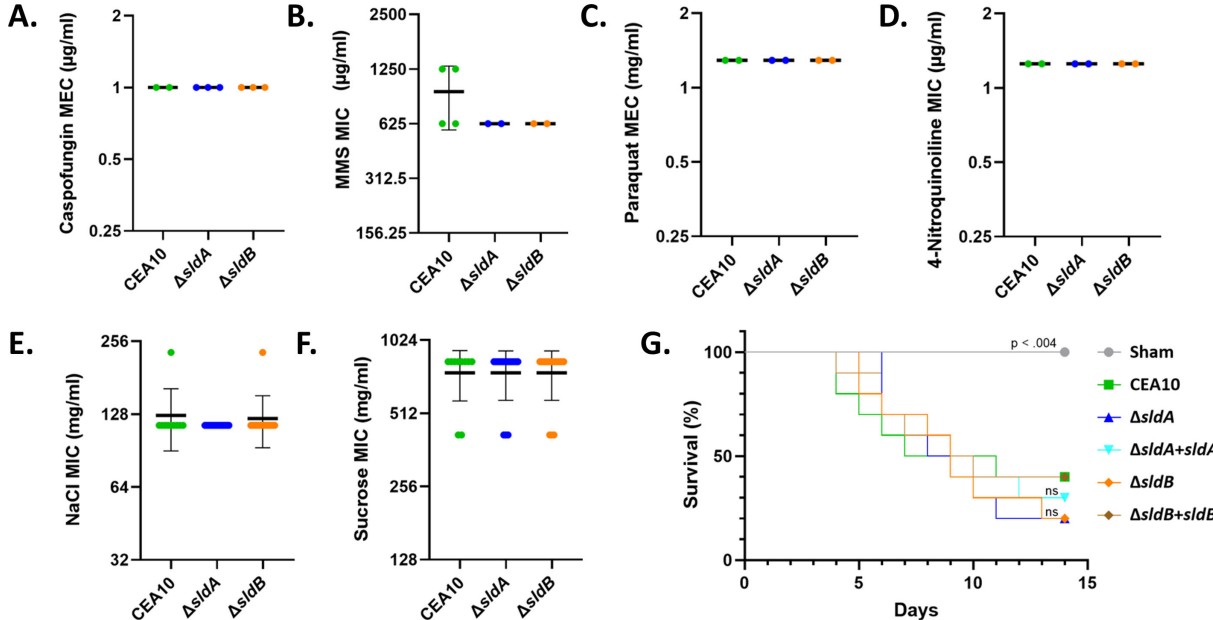

**FIG 6** Loss of the gene *sldA* or *sldB* does not impact susceptibility to cell wall, DNA damage, oxidative, or host stress. MIC/MEC results for select stress-inducing compounds (A) caspofungin, (B) methyl methanesulfonate (MMS), (C) paraquat, (D) 4-nitroquinoline N-oxide, (E) NaCl, and (F) sucrose. The minimum inhibitory concentration (MIC) or minimum effective concentration (MEC) was determined as applicable. Individual results from multiple experiments are shown within each graph. Bold central line denotes the mean MIC. Error bars represent standard deviations. (G) Mice (*n* = 10/group) were chemotherapeutically immune suppressed with both cyclophosphamide and triamcinolone acetonide and intranasally inoculated with $1 \times 10^6$ conidia of the indicated strain. Survival was followed for 14 days post-inoculation. Analysis for significance was assessed using the Mantel-Cox log-rank test.

strains (50). The analysis also produced a fluorescence peak representative of a single haploid nuclear content ($G_1$) peak as well as a smaller but also distinct population with 2n content, which is also known to occur within *A. fumigatus* conidia flow cytometric analysis (50, 52). To determine if deletion of *sldA* or *sldB* impacts the balanced sorting of chromosomal DNA into conidia at baseline, we then analyzed samples of conidia from wild type, Δ*sldA*, and Δ*sldB*. Flow cytometry analysis revealed a clear shift in mean $G_1$ DNA content to the right in both Δ*sldA* and Δ*sldB* conidial populations, indicative of an average skew toward slightly increased mean genome size when compared to the parental population (Fig. 8B). A right-shifted shoulder was also present in the $G_1$ peak fluorescence signal for both mutant strains, again indicating the presence of conidia possessing DNA content which does not exactly match the haploid and diploid levels. Calculation of the DNA index revealed increases for both Δ*sldA* (1.09) and Δ*sldB* (1.10) versus the CEA10 control (set to 1.0) (Fig. 8B). These results indicate the likely existence of subpopulations of Δ*sldA* and Δ*sldB* conidia which were packaged with an abnormal DNA content preferentially skewed toward an increase in genome size, shifting the total mean $G_1$ peak fluorescence of both SAC mutant strains toward the right.

## DISCUSSION

The eukaryotic cell cycle progresses linearly through four distinct phases, G1, S, G2, and M phases, wherein the cell proceeds through mitosis. Several crucial signaling checkpoints exist throughout these stages to ensure errors do not occur (53). The key checkpoint of mitosis is the SAC, often simply referred to as the mitotic checkpoint (54). This is a conserved surveillance system designed to prevent the unbalanced sorting of DNA during nuclear division in eukaryotic species (54, 55). In studies involving eukaryotic species from yeast to humans, defective SAC is consistently associated with chromosome-kinetochore microtubule attachment errors remaining uncorrected, leading to missegregation of chromosomes at anaphase, often resulting in aneuploidy (54, 56). A

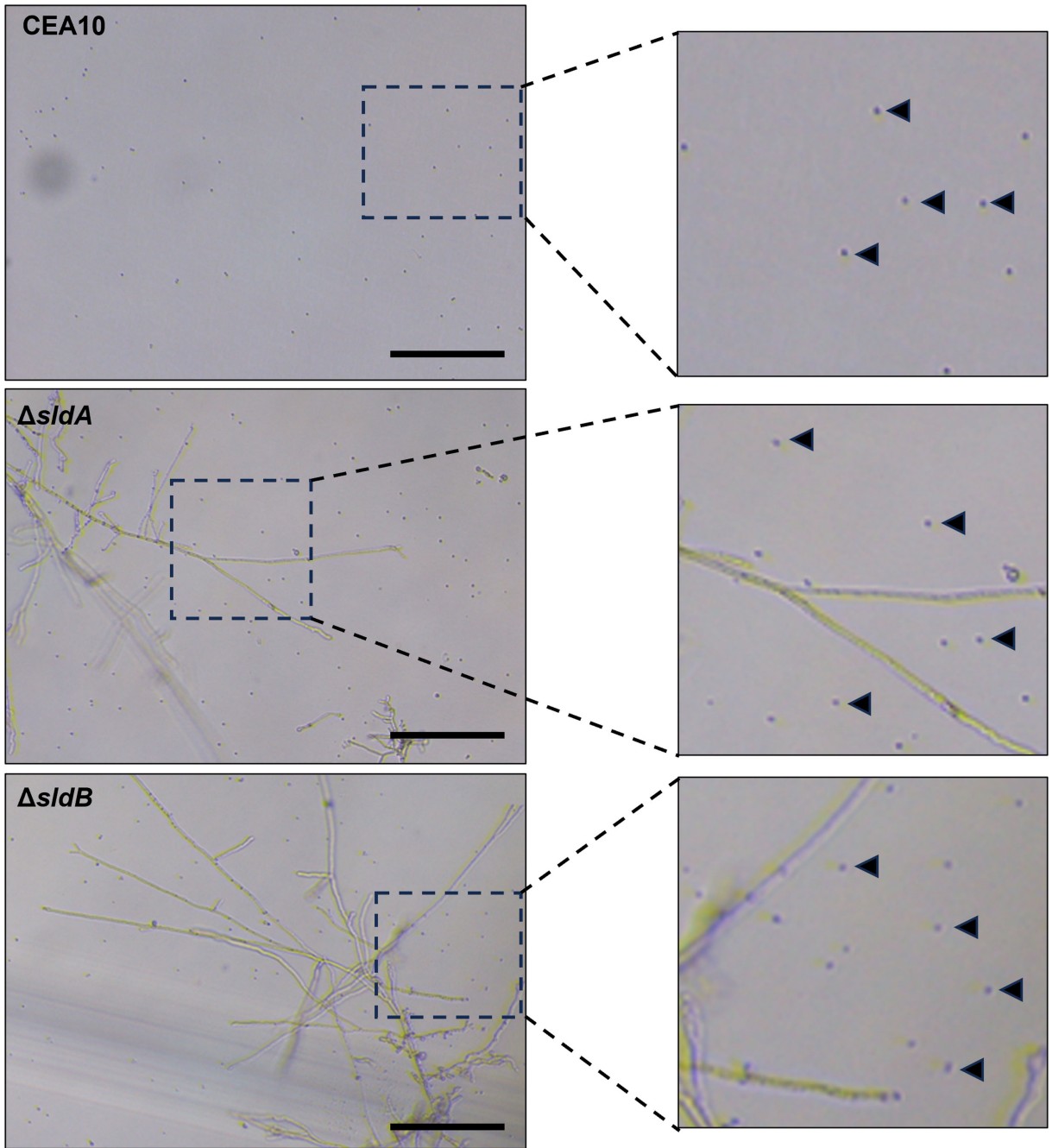

**FIG 7** Loss of *sldA* or *sldB* generates a triazole heteroresistance phenotype. Representative micrographs of the parental strain, CEA10, and the Δ*sldA* and Δ*sldB* mutants acquired from broth microdilution assay wells at 2 µg/mL voriconazole after 48 h incubation at 37°C. Black arrowheads indicate examples of ungerminated conidia after 48 h of culture in the presence of voriconazole. Scale bar = 50 µm.

key protein kinase is required for this checkpoint to occur faithfully and is conserved across eukaryotic species (54, 56). Depending on the species involved, this kinase is known by names such as BUB1 or BUBR1 for "budding uninhibited by benzimidazoles," referring to the characteristic lack of mitotic arrest when the proteins are missing or non-functional despite nocodazole or benomyl treatment; Mad3 for "mitotic arrest deficient"; or SldA for "synthetic lethality with dynein," referring to inviability when the functions of both dynein and the key central SAC kinase are compromised (42, 54). This protein kinase carries out vital activities required to sense the progress of chromosomal alignment and to activate the corrective machinery involved when the requirements of the checkpoint

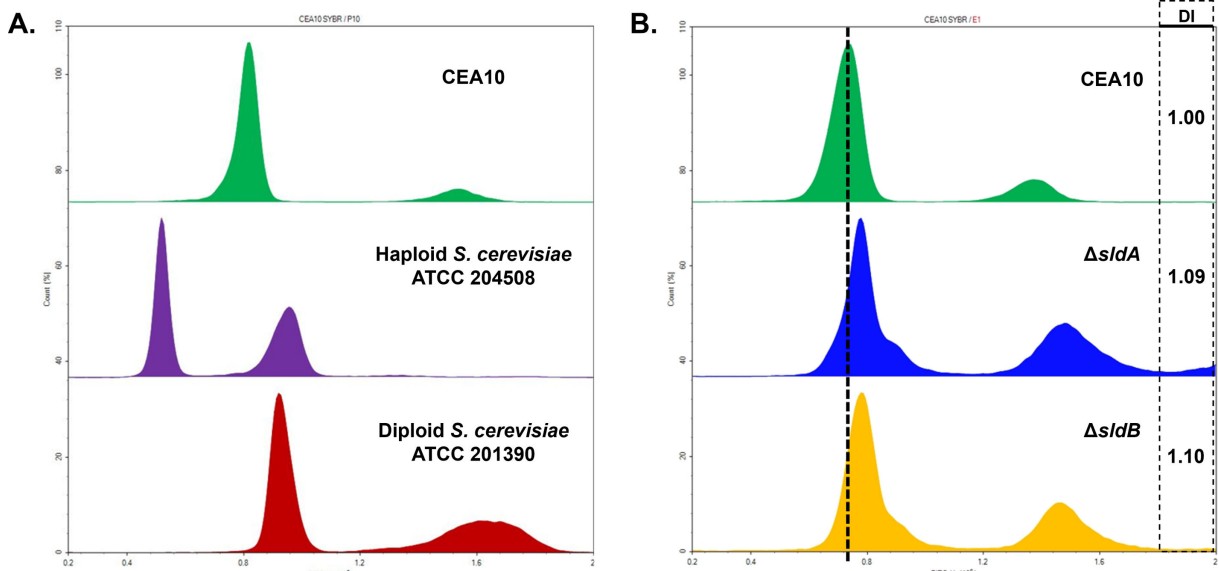

**FIG 8** Conidia of the Δ*sldA* and Δ*sldB* mutants display higher mean DNA content by flow cytometric analysis. (**A**) Yeast cells from both diploid and haploid strains of *S. cerevisiae* were processed alongside wild type CEA10 conidia to detect mean DNA content by flow cytometry. Histogram results from a representative analysis are shown. DNA content is shown on the *x*-axis as measured by fluorescence detected on the fluorescein isothiocyanate (FITC) channel, and normalized count is shown on the *y*-axis. (**B**) Flow cytometric assay comparing the genome content of wild-type CEA10 conidia to that of the Δ*sldA* and the Δ*sldB* deletion strains. The DNA index was calculated based on histogram data and is shown for each strain. A minimum of 70,000 events were measured per sample. DNA content is displayed as the FITC signal on the *x*-axis, with normalized count on the *y*-axis. Dotted line denotes $G_1$ peak fluorescence in CEA10.

are not satisfied (54, 56). Among other roles, this kinase associates with the proteins Mad2, Bub3 (SldB ortholog), and Cdc20 to form a diffusible complex known as the MCC. The MCC inhibits progression to anaphase by inhibiting the association of the anaphase-promoting complex/cyclosome with its coactivator Cdc20 (55). Kinase orthologs also play roles for the SAC in regulating chromosome congression and preserving interchromatid cohesion prior to anaphase (57, 58). Consequently, the function of the kinase is vital to preserve genome integrity.

Prior to the current study, no ortholog of the SldA kinase had been characterized in the filamentous fungal pathogen *A. fumigatus*. However, in species where the orthologous protein kinase has been characterized, important phenotypes are reported, which reveal the importance of these conserved proteins for responding to microtubule errors and safeguarding against abnormal chromosomal sorting during nuclear division. Due to a combination of chromosome congression defects (58, 59), unremedied kinetochore-microtubule attachment errors (58), and premature loss of sister chromatid cohesion (60–64), loss of SAC kinase function or abnormal expression has been shown to result in formation of anaphase bridges, chromosome missegregation, chromosome lagging, and even fragmentation of the DNA when tension is applied across the mitotic spindle. These can result in abnormal levels of genomic DNA within daughter nuclei that receive structurally abnormal chromosomes, micronuclei, and/or the loss or gain of whole or partial chromosomes (58, 59, 63, 65). Organisms deficient in kinase ortholog function are often reported to show hypersensitivity to microtubule-disrupting compounds such as the benzimidazoles benomyl and nocodazole (42, 44). This phenotype is likely due to the compounded stress on the system when two or more factors which support chromosome stability are compromised (42, 65–67). In our study, we observed that loss of either SldA or SldB resulted in a clear hypersusceptibility to benzimidazole and also noted that loss of the SldA kinase resulted in a more profound impact on viability when combined with benzimidazole stress than when the protein SldB was lost. Our results indicate that the functions of these proteins for the SAC are likely conserved in *A. fumigatus* and that loss of the central kinase produces a higher impact on the integrity

of the SAC than does loss of SldB, perhaps due to more severe mitotic defects when combined with spindle microtubule destabilization.

In the present study, we demonstrated that loss of the *A. fumigatus* SAC components SldA or SldB resulted in a triazole heteroresistance phenotype that extended to multiple ergosterol biosynthesis inhibitors. Another recent study also identified multiple protein kinases regulating *A. fumigatus* fitness in response to triazole stress (68). This study by van Rhijn et al. utilized a Bar-seq screen of a signature tagged kinase deletion library in the presence of a sub-MIC voriconazole. Although this study indicated that multiple protein kinases generate increased or decreased fitness when deleted, the sldA deletion mutant was not identified. However, our studies were carried out under different stress conditions and utilized a different endpoint (e.g., standard broth microdilution MIC analyses versus single voriconazole dose Bar-seq) to measure altered susceptibility. We also noted that deletion of either *sldA* or *sldB* generated conidia with altered genome size in the absence of any external stress as detected by fluorescence-activated cell sorting analyses, suggesting that loss of either component partly compromised the fidelity of mitosis during conidial development. Taken together, our findings further support hypotheses presented in recent studies that have demonstrated triazole resistance within *Aspergillus* spp. with altered genome content. In *Aspergillus flavus*, a recent study described the development of aneuploidy within strains that had been experimentally adapted to triazole stress, and in a separate recent report, aneuploidies were detected within triazole-resistant *A. fumigatus* clinical isolates from three patients with invasive and chronic aspergilloses (69, 70). Therefore, it is highly likely that, as with other human pathogenic fungi, aneuploidy is an important mechanism for *Aspergilli* to survive triazole stress. In human pathogenic *Candida* and *Cryptococcus* spp., induction of aneuploidy is an established source for triazole adaptation and acquisition of triazole resistance (47, 71–74). For example, isolates which have increased representation of the left arm of chromosome 5, which is known to contain the genes encoding the target of triazoles in *Candida* spp., ERG11, and the transcription factor TAC1, which upregulates drug efflux pump gene expression, can provide resistance to fluconazole (75). Although similar aneuploidies may be present in our SAC deletion strains, we did not find an increased basal or stress-induced expression of known resistance-associated genes in the Δ*sldA* mutant. However, we must note the caveat that our expression studies were conducted after a short time course of drug exposure (i.e., 4 h) using mature hyphae grown under non-stress conditions as the starting material, whereas our heteroresistance phenotype is recorded at a later timepoint (i.e., 48 h of drug exposure) beginning with the conidial phase. In studies of the human pathogenic fungal yeast *C. albicans*, loss of either SAC/MCC component, BUB1 kinase, or Mad2 results in increased baseline aneuploidy and reduced triazole susceptibility (72, 76). Moreover, the rate of aneuploidy increases further for these mutant strains in the context of fluconazole stress (76). In fact, isolates of *C. albicans* have been identified that have acquired a natural adaptation which leverages SAC failure via Bub1-phosphoregulation deficiencies to enhance both aneuploidy generation and reduce fluconazole susceptibility (72). These isolates were found to possess a variant of histone H2A, a Bub1 kinase target, in which the site of BUB1-mediated phosphorylation is mutated, generating a version of histone H2A which cannot be regulated by the SAC kinase. Strains which express the H2A variant produce chromosome missegregation that enables accumulation of known triazole resistance-conferring chromosome 5 duplications at a rate six times higher than the wild-type comparator and five times higher than wild-type *C. albicans* in the context of fluconazole exposure. This is a similar level of reduced fluconazole susceptibility as noted in an SAC-deficient BUB1 kinase deletion strain (72). Similar histone variants were also found to exist in other *Candida* spp., including *Candida tropicalis* and *Candida dubliniensis*, indicating that the ability to disrupt SAC fidelity in an inducible manner is retained in these species as well (72). Moreover, the same study found that *C. albicans* could also promote fluconazole adaptation by regulated depletion of Cse4 (CENP-A), a variant of the centromere-specific histone H3, which interacts with the Bub1-target histone H2A

and forms part of the platform onto which the kinetochore is built in eukaryotes (72, 77). Defects in H2A or CENP-A function or expression are known to induce CIN and aneuploidy in species from the fungi *S. cerevisiae* and *Schizosaccharomyces pombe* (77, 78) to humans, mice, and *Drosophila melanogaster* (79–82). Loss of triazole susceptibility has also been previously described due to defects in the SAC-regulating component Aurora B in *C. neoformans* (83). Thus, abnormal function or altered expression of SAC components has the potential to provide chromosome abnormalities and reduced triazole susceptibility in fungal pathogens.

Taken together, this work further informs our understanding of mechanisms underpinning triazole susceptibility in *A. fumigatus*. Future studies will explore if SAC component deletion induces the development of aneuploidies in *A. fumigatus* by performing copy number variation (CNV) analyses and will work to identify if specific alleles are duplicated to drive resistance to triazoles. If aneuploidy occurs in the SAC mutants, it will be informative to define how triazole stress impacts the frequency and nature of specific CNVs and whether aneuploidy in an adapted strain is stable or transient. These are essential questions to answer to define the role of such potential aneuploidies to support the eventual development of genetically encoded resistance in *A. fumigatus*. As our *A. fumigatus* SAC mutants are resistant to a variety of ergosterol biosynthesis inhibitors, additional work is also required to decipher if specific connections exist between ergosterol biosynthesis and mitotic fidelity.

## MATERIALS AND METHODS

### Strains and growth conditions

The wild-type strain CEA10/A1163 was used as the background strain for all *Aspergillus fumigatus* genetic manipulations in this study. The *Aspergillus nidulans* wild-type strain, FGSC A4, was acquired from the Fungal Genetics Stock Center. All strains were propagated on *Aspergillus* glucose minimal medium (GMM) agar at 37°C (84). Submerged culture and broth microdilution assays were performed using either Roswell Park Memorial Institute 1640 medium or GMM broth as indicated.

### Genetic manipulations of *A. fumigatus*

All primers and CRISPR/Cas9 gene-editing components used in this study are listed in Table 1. Genetic manipulations described in this study were accomplished using a modified CRISPR-Cas9 technique, which has been described previously (41). For the generation of gene deletion mutants, the entire open reading frame (ORF) was replaced by a homologous repair template containing the hygromycin resistance cassette. The genetic sequence encoding SldA and SldB in *A. fumigatus* was identified through BLAST search using verified gene sequences from the model organism *A. nidulans* (39, 42, 44). Optimal protospacer adjacent motif (PAM) sites were identified upstream and downstream of the gene of interest using the Eukaryotic Pathogen CRISPR guide RNA/DNA Design Tool made available online by the University of Georgia (85). As complete gene deletion requires induction of a double-strand break (DSB) both upstream and downstream of the gene coding sequence within the DNA, we selected two optimal PAM sites and designed the repair template to contain the gene encoding the hygromycin resistance cassette amplified from the plasmid pJMR2 (30), as well as the 40 bp microhomology upstream of the 5′ DSB and to 40 bp downstream of the 3′ PAM site. The homology repair templates were generated by PCR using high-fidelity TAQ polymerase master mix. *In vitro* assembly of the ribonucleoprotein complexes, protoplast generation, and transformations were performed as previously described (41). For fungal transformations, GMM supplemented with 1.2 M sorbitol (sorbitol minimum medium) was utilized to allow the protoplasts to adequately recover cell wall stability prior to selection. Hygromycin-resistant colonies were screened and genotyped by PCR to confirm correct integration into the genome (86). For complementation of the deletion mutants, the entire ORF was reinserted into the genome at its native locus, adding the phleomycin

**TABLE 1** Primers and CRISPR/Cas9 components used in this study

| Primer/crRNA Name | 5′–3′ sequence | Description of use |
|---|---|---|
| crRNA Δ*sldA* 5′ | TTGTTTATCTGTCGCCGCCA | *sldA* deletion |
| crRNA Δ*sldA* 3′ | CTGCCGGAGGAGGAGAAACAA | *sldA* deletion |
| crRNA Δ*sldB* 5′ | CATGTCAAGGCGGTATACGC | *sldB* deletion |
| crRNA Δ*sldB* 3′ | CCGCGGCTGAGAATCCAGAA | *sldB* deletion |
| crRNA Δ*sldA* + *sldA* | AAACAAGCGTGCGGTCCAAA | Δ*sldA* complementation |
| crRNA Δ*sldB* + *sldB* | GTTGACTTCCCCGCACCCGG | Δ*sldB* complementation |
| Primer *sldA1* | CCCTTTGGACCGCACGCTTGTTTATCTGTCGCCGCCATTGAGCTTGCATGCTGCAGGTCGAGTGGAGATGTGGAGTGGG | Δ*sldA* repair template |
| Primer *sldA2* | AGATCCAAACGCCAACGCTGCCGGAGGAGGAGAAACAATGTCGAGCTCCCAAATCTGTCCAGATCATGGTTGACCGGTGCC | Δ*sldA* repair template |
| Primer *sldA3* | GGAGGCGGAGAATCCTAGTTTGACG | Δ*sldA* screen |
| Primer *sldA4* | CCGTCCTCCAAAATTACCAAGAGGAGG | Δ*sldA* screen |
| Primer *sldA5* | TTGAGATATCAGACTCCGACACGAGCGC | Δ*sldA* screen |
| Primer *sldB1* | GATGGCGGGGAGGGAAACATGTCAAGGCGGTATACGCAGTAGCTTGCATGCCTGCAGGTCGAGTGGAGATGTGGAGTGGG | Δ*sldB* repair template |
| Primer *sldB2* | CTTGCCGGTCGAATCTCACCGGCGCTGAGAATCCAGAATGTCGAGCTCCCAAATCTGTCCAGATCATGGTTGACCGGTGCCT | Δ*sldB* repair template |
| Primer *sldB3* | TTTGCATGTGATAATCGTGATGACTCGGCC | Δ*sldB* screen |
| Primer *sldB4* | TTAGTGAGGACGTTGATGTGAGGGAGATCC | Δ*sldB* screen |
| Primer *sldB5* | TATATTCCCATTGAGGTGCACCTATTGCC | Δ*sldB* screen |
| Primer *HygR1* | TCGCCGATAGTGGAAACCGACGC | Δ*sldA* and Δ*sldB* screen |
| Primer *HygR2* | TTAGGCTCAAGTCATGACCCTCTGG | Δ*sldA* and Δ*sldB* screen |
| Primer *HygR3* | TCGTGTACTGTGTAAGCGCC | Δ*sldA* and Δ*sldB* screen |
| Primer *HygR4* | TTTCGGGAGACGAGATCAAGCAG | Δ*sldA* and Δ*sldB* screen |
| Primer Δ*sldA* + *sldA1* | GTTGCCCTCGACAAATCATTTCACGACTCACCAACAGACTATGGCGGCCTCCGAGGATCTCATCAATTTT | Δ*sldA* complementation |
| Primer Δ*sldA* + *sldA2* | GAAGGTTTTGGGACGCTCGAAGGCTTTAAATTCAGCTCCGTTCCAGTTTCTTCTTCTTCTCG | Δ*sldA* complementation |
| Primer Δ*sldA* + *sldA3* | CGAGAAGAAGAAGAAACTGAACGGAGCTGAAATTAAAGCCTTCGAGCGTCCCAAAACCTTC | Δ*sldA* complementation |
| Primer Δ*sldA* + *sldA4* | GCTTGGCGGCGACAGATAAACAAGCGTGCGGTCCAAAGTGAAGCTTCGGAGAATATGGAGCTTCATCGAATC | Δ*sldA* complementation |
| Primer Δ*sldA* + *sldA5* | TTCTTCCTCTACTAGAACGCGCCGTCAG | Δ*sldA* complementation screen |
| Primer Δ*sldA* + *sldA6* | TTGTGCAGAGGGGTAAGCGTTCAGTG | Δ*sldA* complementation screen |
| Primer Δ*sldA* + *sldA7* | TTGGCTCAATGCGCGAGAGACGTAAG | Δ*sldA* complementation screen |
| Primer Δ*sldA* + *sldA8* | TTGAACGAGAGCAAAGCTAACAAAGCACGG | Δ*sldA* complementation screen |
| Primer Δ*sldB* + *sldB1* | GCATACCGTTGCAATATGTTGACTTCCCCGCACCCGGGtATGGCATCAAGTAAGGTCAATAGAAACCGC | Δ*sldB* complementation |
| Primer Δ*sldB* + *sldB2* | TTTTGGGACGCTCGAAGGCTTTAATTTTATTTCGCACCCTTTCCTTTCGCCTCAGTCTC | Δ*sldB* complementation |
| Primer Δ*sldB* + *sldB3* | TGGGCGAGACTGAGGCGAAAGGAAAAGGGTGCGAAATAAAATTAAAGCCTTCGAGCGTCCC | Δ*sldB* complementation |
| Primer Δ*sldB* + *sldB4* | CGCTCGCAGCTTAAAATGCCTTTCGCGCAGCTGGTAATAGAAGCTTCGGAGAATATGGAGCTTCATCGAATC | Δ*sldB* complementation |
| Primer Δ*sldB* + *sldB5* | CAGCAAGGAGCATAACATTGTCATATCTGCG | Δ*sldB* complementation screen |
| Primer Δ*sldB* + *sldB6* | TCGATATCAACCTTGTTGGGAGATGTGGCG | Δ*sldB* complementation screen |
| Primer Δ*sldB* + *sldB7* | CTGTTAGCCCTGGTTTTGAAGACGGGAAAAG | Δ*sldB* complementation screen |

resistance cassette. Complementation required two repair template fragments to be made via PCR amplification and product purification. For the first fragment, we amplified the complete ORF for the gene of interest from wild-type CEA10 using primers adding 40 bp of homology to the 5′ DSB and 40 bp homology to the phleomycin resistance sequence. We produced the second fragment by amplifying the phleomycin resistance repair cassette from the plasmid pAGRP (87) using primers adding 40 bp homology to the 3′ end of the gene of interest ORF and 40 bp homology to the 3′ PAM site in the genome. Consequently, the repair template fragments possessed overlapping regions which recombined during the transformation process by the endogenous repair machinery. The resulting gene transformation recapitulated the original sequence of the gene in its 5′ context while incorporating the phleomycin resistance selection cassette downstream of the gene of interest and upstream of the hygromycin cassette. Positive colonies resistant to both hygromycin and phleomycin were confirmed by multiple genotyping PCR reactions to verify correct integration of the homologous repair template (86).

## Assays for analysis of growth and stress resistance

Analysis of radial growth rate was performed for each strain following a previously described protocol (88). Briefly, 5 µL of a $10^6$ cfu/mL water stock was spot-inoculated into the center of a 100 mm GMM agar plate and incubated at 37°C. The colony diameter for each strain was measured at 24 h intervals, and images of plates were captured at 96 h of incubation for visual representation of colony formation on solid medium. Determination of biomass accumulation was performed by inoculating culture tubes containing 5 mL of GMM supplemented with 0.5% yeast extract (GMM + YE) (86) to a final concentration of $10^5$ cfu/mL and incubating at 37°C for 24 or 48 h with shaking at 250 rpm. After the indicated timepoints, the mycelium was harvested, dried, and lyophilized for a minimum of 24 h before the final dry weight was recorded. Germination assays were performed following a previously outlined protocol, with modification (89). Briefly, for each strain, coverslip cultures containing GMM broth were inoculated to a final concentration of $10^5$ cfu/mL and incubated at 37°C. Coverslips for each strain were removed from the cultures at the times indicated, and the number of conidia with a visible germ tube was counted. For these analyses, each experiment was performed at minimum in triplicate, and data are reported as mean ± standard deviation. Analysis for statistically significant differences between the mutant and wild type CEA10 groups was performed via one-way analysis of variance with Tukey's or Dunnett's post hoc test as noted in the figure (GraphPad Prism v.9.5.1.).

Spot dilution assays to determine susceptibility of strains were performed as previously described, with modifications (37). Briefly, fresh conidial suspensions were prepared for each control and test strain. Assays were accomplished using 100 mm round petri plates containing 20 mL of GMM with or without the appropriate final concentration of specified compound added homogenously into the medium. For each strain, 10 µL of 10-fold serial dilution of conidia ranging from $5 \times 10^6$ to $5 \times 10^3$ conidia/mL was spot-inoculated onto GMM agar plates supplemented with the concentrations of compound indicated. GMM agar plates without drug were included in each experiment as a growth control. Plates were incubated at 37°C for the timepoints indicated within each figure, at which point colony growth was analyzed and images were captured.

The susceptibility profiles of stress-inducing compounds referenced in this study were also determined for each strain by broth microdilution assays performed as previously described, with modification (37). Briefly, 10 twofold dilutions of the appropriate compound were prepared in GMM broth within 96-well plates. Fresh conidial stock suspensions were prepared, and each well was inoculated with a total of $2 \times 10^4$ conidia from the appropriate test or control strain. Control wells including no conidia or no drug were included for each row within the first two wells. Plates were incubated at 35°C for 24 h for echinocandin MEC (37) or at 37°C for 48 h for other compounds. After the appropriate incubation, the MIC or MEC was determined as applicable, dependent

on the compound used in the experiment. MIC was considered the concentration of the compound which inhibited 100% of growth. For determining the susceptibility of *Aspergillus nidulans* wild-type and mutant strains to voriconazole by broth microdilution assay, 0.5% yeast extract was added to the medium (GMM + YE) to encourage sufficient growth within the 48 h timeframe.

## Fluorescent staining and microscopy

PI and Calcofluor White co-staining was performed as previously described (37). Sterilized coverslips were submerged in 5 mL of GMM broth and inoculated with $1 \times 10^3$ conidia of the appropriate strain. Coverslip cultures were incubated at 37°C. At the specified timepoints, coverslips with adherent hyphae were removed from incubation and washed with 50 mM morpholinepropanesulfonic acid (MOPS) buffer adjusted to pH 6.7. Washed coverslips were then submerged in a fixative solution of 8% formalin, 25 mM EGTA, 5 mM MgSO$_4$, 5% dimethyl sulfoxide, and 0.2% Triton X for 1 h at room temperature (RT). Coverslips with fixed adherent hyphae were then washed twice for 10 min with a 50 mM solution of piperazine-N,N′-bis(2-ethanesulfonic acid) (PIPES), then treated for 1 h at 37°C with RNAse A at a final concentration of 100 µg/mL prepared in the PIPES buffer. Coverslips were then washed twice with the MOPS buffer. Following the second wash, coverslips were stained with a solution of 12.5 µg/mL of propidium iodide and 1 µg/mL of Calcofluor White in a light-proof container for 5 min at RT. Coverslips were then washed twice with MOPS before being mounted onto glass microscope slides. Stained coverslips were analyzed immediately after being mounted to capture images of sufficient number and quality for analysis. Images of stained adherent hyphae were obtained using a Nikon NI-U upright fluorescence microscope equipped with both tetramethylrhodamine isothiocyanate and 4′,6-diamidino-2-phenylindole filters and with use of the Nikon Elements software package.

## Assessment of differential expression of triazole resistance-associated genes

Real-time quantitative PCR (RT-qPCR) assays to measure gene expression were performed as previously described, with slight modifications (37). Cultures were prepared for total RNA extraction by inoculating $2 \times 10^7$ total conidia into GMM + YE. For the voriconazole-treated experimental groups, voriconazole was added to the medium at a final concentration of 0.5 µg/mL. Cultures were then incubated at 37°C at 250 rpm. At the indicated timepoints, cultures for each strain were removed from the incubator and placed on ice until processing. Each sample was processed for extraction of total RNA using the Trizol extraction method and cDNA synthesized using the ProtoScript II First Strand cDNA synthesis kit (New England Biolabs). Normalized dilutions of cDNA were added to iQ SYBR Green Supermix (2×) (Bio-Rad) combined with gene-specific forward and reverse primers for RT-qPCR reactions. Relative fold expression of the genes encoding the *A. fumigatus* 14α-lanosterol demethylase enzymes (Cyp51A and Cyp51B), the triazole-resistance-associated efflux pumps (AbcC/Cdr1B and AtrF), and *sldA* was determined using the $2^{-\Delta\Delta Ct}$ method, as previously described (90).

## Flow cytometry analyses and calculation of DNA index

Flow cytometric quantitation of DNA content was performed as previously described (50, 91). In brief, samples of conidia from each test and control strain were washed once with sterile 1× phosphate-buffered saline (PBS), resuspended in PBS, and filtered through sterile miracloth to remove clumped conidia. Washed conidia were then resuspended in a 70% concentration (vol/vol) solution of EtOH and fixed overnight at 4°C with agitation. Following ethanol fixation, the samples of conidia were washed twice with sterile PBS, then resuspended in 50 mM sodium citrate buffer (pH 7.5) and sonicated for three 1 second pulses separated by a 2 second pause between each pulse. Following sonication, RNase A (Invitrogen, Waltham, MA, USA) was added to each sample to a final concentration of 0.50 mg/mL, and samples were incubated for at least 1 h at 50°C.

After incubation, proteinase K (Sigma-Aldrich, St. Louis, MO, USA) was added to each sample to a final concentration of 1 mg/mL, and samples were incubated for at least 2 h at 50°C. Conidia were then washed twice with sterile PBS and again resuspended in 50 mM sodium citrate buffer. Samples were then sonicated as previously described. Samples were then divided between two tubes; Tube 1 was stained overnight with SYBR Green DNA stain (Millipore Sigma), and Tube 2 was unstained as a control for baseline fluorescence for flow cytometry analyses. For the tubes of conidia to undergo staining, $10,000\times$ SYBR Green was diluted in Tris-EDTA buffer (pH 8.0) and added to each sample of resuspended, sonicated conidia to a final concentration of 2% (vol/vol). After the addition of the stain, sample tubes were incubated for 16 h at 4°C with agitation in a light-proof container. Following staining, both stained and unstained samples of conidia were washed twice in sterile PBS. Washed conidia were resuspended in PBS with 0.25% (vol/vol) Triton X added to concentrations of approximately $1 \times 10^7$ conidia/mL and transferred into tubes compatible for use with the flow cytometer. For each SYBR-stained conidial sample, an equivalent sample of unstained conidia was included as a control for baseline fluorescence. Samples were analyzed by flow cytometry using a NovoCyte 3000 unit (92). The cytometer was set to a low flow rate of ~1,000 cells/second, and a minimum of 70,000 events were captured using forward scatter and side scatter gating strategies previously optimized for *A. fumigatus* conidia. Measures of fluorescence were obtained using a 488 nm laser for excitation and the fluorescein isothiocyanate channel (530 ± 30 nm) for detection of emission. DNA index was calculated following previously published protocols (93). In brief, the mean $G_1$ peak fluorescence was first identified for the wild-type CEA10 sample. Then, the mean $G_1$ peak fluorescence was determined for each experimental sample. Finally, the mean $G_1$ peak fluorescence of the test sample was divided by the mean $G_1$ peak fluorescence of the control CEAG1e same analysis, resulting in the DNA index score for the sample.

## Murine model of invasive aspergillosis

The animal model of invasive pulmonary aspergillosis was employed as previously described (37). Six-week-old female CF-1 mice weighing approximately 25 g were immunosuppressed by subcutaneous injection of 40 mg/kg triamcinolone acetonide (Kenalog; Bristol-Meyers Squibb, Princeton, NJ, USA) given 1 day prior to infection (Day −1). Mice were also immunosuppressed by intraperitoneal injection of 150 mg/kg cyclophosphamide administered every 3 days for the duration of the experiment, beginning on Day −3. On Day 0, mice were transiently anesthetized by application of inhaled isoflurane within an induction chamber with the primary and secondary flow rates set to 0.5 L/min and 2.5% isoflurane. Sedated mice were then inoculated by intranasal instillation of a total of $1 \times 10^6$ conidia, suspended within 20 µL of sterile saline solution. The health of each mouse was monitored at least twice daily for the duration of the experiment. Survival was recorded for 14 days post-inoculation. Mice which reached the criteria requiring humane euthanasia, including signs of distress and end stages of disease, were humanely euthanized by anoxia with $CO_2$ followed by cervical dislocation. Statistical analysis for differences in survival between the no-conidia control, CEA10, and mutant strain infected groups was performed using the Mantel-Cox log-rank test (37).

## ACKNOWLEDGMENTS

This work was supported by the National Institutes of Health/National Institute of Allergy and Infectious Diseases (grants R01AI143197 and R01AI158442, awarded to J.R.F.).

## AUTHOR AFFILIATIONS

[1]Department of Clinical Pharmacy and Translational Science, College of Pharmacy, University of Tennessee Health Science Center, Memphis, Tennessee, USA
[2]College of Graduate Health Sciences, Integrated Biomedical Sciences Program, University of Tennessee Health Science Center, Memphis, Tennessee, USA

³Graduate Program in Pharmaceutical Sciences, College of Pharmacy, University of Tennessee Health Science Center, Memphis, Tennessee, USA

⁴Department of Pharmacy and Pharmaceutical Science, St Jude Children's Research Hospital, Memphis, Tennessee, USA

⁵Department of Food Science, College of Natural Sciences, University of Massachusetts Amherst, Amherst, Massachusetts, USA

## AUTHOR ORCIDs

Adela Martin-Vicente ⓘ http://orcid.org/0000-0003-0446-8906
John G. Gibbons ⓘ http://orcid.org/0000-0003-1282-9345
Jarrod R. Fortwendel ⓘ http://orcid.org/0000-0003-2301-4272

## FUNDING

| Funder | Grant(s) | Author(s) |
|---|---|---|
| National Institute of Allergy and Infectious Diseases | R01 AI143197, R01 AI158442 | Jarrod R. Fortwendel |

## AUTHOR CONTRIBUTIONS

Ashley V. Nywening, Conceptualization, Data curation, Formal analysis, Investigation, Methodology, Writing – original draft, Writing – review and editing | Harrison I. Thorn, Data curation, Investigation, Methodology, Writing – review and editing | Jinhong Xie, Data curation, Formal analysis, Investigation, Methodology, Writing – review and editing | Adela Martin-Vicente, Data curation, Investigation, Methodology, Writing – review and editing | Xabier Guruceaga, Data curation, Investigation, Methodology, Writing – review and editing | Wenbo Ge, Data curation | John G. Gibbons, Data curation, Formal analysis, Writing – original draft | Jarrod R. Fortwendel, Conceptualization, Data curation, Formal analysis, Funding acquisition, Project administration, Supervision, Writing – original draft, Writing – review and editing

## ETHICS APPROVAL

All animal studies were carried out with the approval of the University of Tennessee Health Science Center Institutional Animal Care and Use Committee under protocol 22-0373.0.

## ADDITIONAL FILES

The following material is available online.

### Supplemental Material

**Fig. S1 (Spectrum00536-25-S0001.tif).** Schematics for gene targeting.
**Fig. S2 (Spectrum00536-25-S0002.tif).** Triazole MIC of *Aspergillus nidulans* sldA mutant.
**Fig. S3 (Spectrum00536-25-S0003.tif).** Gene expression analyses.
**Supplemental material (Spectrum00536-25-S0004.docx).** Legends for all Supplemental figures.

### Open Peer Review

**PEER REVIEW HISTORY (review-history.pdf).** An accounting of the reviewer comments and feedback.

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
