## [Reviewer comments · Microbiology Spectrum]

Microbiology Spectrum

Loss of the *Aspergillus fumigatus* Spindle Assembly Checkpoint components, SldA or SldB, generates triazole heteroresistant conidial populations

Ashley Nywening, Harrison Thorn, Jinhong Xie, Adela Martin-Vicente, Xabier Guruceaga, Wenbo Ge, John Gibbons, and Jarrod Fortwendel

Corresponding Author(s): Jarrod Fortwendel, The University of Tennessee Health Science Center College of Pharmacy

Review Timeline:

Submission Date:	February 22, 2025
Editorial Decision:	March 11, 2025
Revision Received:	May 12, 2025
Accepted:	May 19, 2025

Editor: James Konopka

Reviewer(s): Disclosure of reviewer identity is with reference to reviewer comments included in decision letter(s). The following individuals involved in review of your submission have agreed to reveal their identity: Gustavo H. Goldman (Reviewer #1); W. Scott Moye-Rowley (Reviewer #2)

Transaction Report:

DOI: <https://doi.org/10.1128/spectrum.00536-25>

Re: Spectrum00536-25 (**Loss of the *Aspergillus fumigatus* Spindle Assembly Checkpoint components, SldA or SldB, generates triazole heteroresistant conidial populations**)

Dear Dr. Jarrod R. Fortwendel:

Thank you for the privilege of reviewing your work. Below you will find my comments, instructions from the Spectrum editorial office, and the reviewer comments.

The reviewers agreed that your manuscript addresses an important topic - new mechanisms that promote azole drug resistance in *Aspergillus fumigatus*. Overall, it seems that the reviewers thought the presentation of data in the manuscript was clear. However, both reviewers raised questions about the strength of the data supporting the link between aneuploidy and azole resistance. It would therefore strengthen the manuscript if you could provide additional support for this conclusion and adjust the discussion of the results accordingly.

Revision Guidelines

Sincerely,
James Konopka
Editor
Microbiology Spectrum

Reviewer #1 (Comments for the Author):

This manuscript describes a screening of disruption mutants of *Aspergillus fumigatus* for azole-susceptibility. It is an interesting manuscript that touches again an evidence that progressively is being introduced in the literature, i.e., the importance of aneuploidy for azole-resistance. I would like to suggest a couple of modifications that could strength the observations provided by the authors:

- 1) Lines 43 to 48: It is mentioned "production of aneuploid conidia" and "link exists between aneuploidy development and resistance to ergosterol biosynthesis perturbation in *A. fumigatus*". Is it enough the simple peaks detected in the FACs as a prove for aneuploidy ?
- 2) Lines 118 to 120: Have you screened for resistance and sensitivity ? Please, clarify.
- 3) Line 267: nad ?
- 4) Please, compare and discuss your findings with a recent paper published by Bromley's lab (10.1038/s41467-024-48592-8)
- 5) Flow cytometry coupled with additional molecular methods such as comparative genome hybridization or genome sequencing could be used as stronger evidences for aneuploidy
- 6) Is it possible to enrich the aneuploids by incubating the mutants in the presence of azoles (several transfers) and checking the frequency of heteroresistance versus the enrichment of the specific peaks at FACs ? Also looking microscopically this material at the frequency of germinated conidia could be a good indicator
- 7) Could the authors speculate if this mechanism of aneuploid created by heteroresistance can lead to azole-resistance acquisition, and if so, aneuploid is going to be maintained or it is only a temporary process before the mutated alleles that confer resistance are established ?

Reviewer #2 (Comments for the Author):

This manuscript reports the result from screening a collection of kinase disruption mutations in *Aspergillus fumigatus* (Afu) for potential regulators of resistance to azole drugs. Surprisingly, of the 118 null mutant strains constructed and screened for azole susceptibility, only two were identified that had MICs at least 4-fold different from that of the wild-type strain. The second unexpected result was that these two mutants strains both exhibited decreased susceptibility to voriconazole compare to the wild-type strain. One disruption corresponded to the Afu *ssn3* gene and was previously reported to be less susceptible to azole challenge, validating the screening approach. The second gene, and the focus of this work, represented the Afu homologue of the Spindle Assembly Checkpoint kinase *SldA*.

To explore the role of *SldA* in development of decreased azole susceptibility, the authors generated a new complete disruption allele of this gene and the potential *SldA*-binding protein gene: *sldB*. Null alleles of either gene had no major growth defects, did exhibit increased benomyl susceptibility but were not essential for nuclear distribution through Afu hyphae. Both null allele-containing strains exhibited decreased susceptibility to other sterol biosynthesis inhibitors but showed normal amphotericin B susceptibility. Transcriptional analyses of several key genes involved in azole susceptibility failed to show any altered expression of these loci, consistent with a different mechanism of azole resistance in *sldA/B* null strains. Phenotype testing with a range of other stress agents failed to reveal any other alterations in *sldA/B* growth. The authors suggest a heteroresistance phenotype linked with aneuploidy being caused by loss of either gene and support this view with a demonstration of altered DNA content of each mutant compared to wild-type.

This is a well-done piece of work that will be useful for investigators studying antifungal drug resistance in Afu and other filamentous fungi. Much of what is shown here had already been observed in other fungi (as the authors point out) but these are new data specifically in Afu and filamentous fungi in general.

I only have one minor point that I think the authors should address in the discussion. While aneuploidy certainly seems like the most likely cause of the decreased azole susceptibility, this is never firmly established. The lack of an effect on transcription of azole resistance genes may not be seen here because the time point used (4 hours) is quite short compared to the phenotypic demonstration of the decreased azole susceptibility. It seems likely that altered genomic content could lead to increased expression of relevant target genes and the failure to see this here may just reflect differences between the development of the aneuploidy and the timing used here.

Response to Reviewer Comments

Reviewer #1:

Comment: 1) Lines 43 to 48: It is mentioned "production of aneuploid conidia" and "link exists between aneuploidy development and resistance to ergosterol biosynthesis perturbation in *A. fumigatus*". Is it enough the simple peaks detected in the FACs as a prove for aneuploidy?

Response: We appreciate the reviewer's comment here and agree that the data we have provided is supportive, but certainly not conclusive, concerning the issue of aneuploidy. We agree that the additional experimentation proposed by the reviewer in Comments 5 and 6 would help to better establish aneuploidy in our mutant strains. While studies using repeated exposure and Copy Number Variation analyses are ongoing in our lab, they will require a prolonged experimental timeline. We plan to publish that work in the future as part of a larger study exploring triazole stress-induced aneuploidy and its consequences to susceptibility. We have therefore decided to soften our language throughout this manuscript regarding aneuploidy.

Comment: 2) Lines 118 to 120: Have you screened for resistance and sensitivity? Please, clarify.

Response: Yes, both were measured and the reason why we used the phrase "alterations" in voriconazole susceptibility rather than "increase" or "decrease". We have edited the text to clarify further.

Comment: 3) Line 267: nad?

Response: Typo. We have edited this to "and"

Comment: 4) Please, compare and discuss your findings with a recent paper published by Bromley's lab (10.1038/s41467-024-48592-8)

Response: While this manuscript mainly focuses on a kinase that is involved in septal plugging in response to environmental stress and is therefore required for cell wall stress, the authors do perform an initial fitness-based screen of a kinase deletion library to identify changes in voriconazole susceptibility. We have now included this in the Discussion section of the paper.

Comment: 5) Flow cytometry coupled with additional molecular methods such as comparative genome hybridization or genome sequencing could be used as stronger evidences for aneuploidy

Response: Please see our response to Comment #1.

Comment: 6) Is it possible to enrich the aneuploids by incubating the mutants in the presence of azoles (several transfers) and checking the frequency of heteroresistance versus the enrichment of the specific peaks at FACs? Also looking microscopically this material at the frequency of germinated conidia could be a good indicator

Response: Please see our response to Comment #1.

Comment: 7) Could the authors speculate if this mechanism of aneuploid created by heteroresistance can lead to azole-resistance acquisition, and if so, aneuploid is going to be maintained or it is only a temporary process before the mutated alleles that confer resistance are established?

Response: We thank the reviewer for this insightful thought and have updated the Discussion to speculate on the potential for aneuploidy to generate encoded resistance and if it is lost or maintained in the process.

Reviewer #2:

Comment: I only have one minor point that I think the authors should address in the discussion. While aneuploidy certainly seems like the most likely cause of the decreased azole susceptibility, this is never firmly established. The lack of an effect on transcription of azole resistance genes may not be seen here because the time point used (4 hours) is quite short compared to the phenotypic demonstration of the decreased azole susceptibility. It seems likely that altered genomic content could lead to increased expression of relevant target genes and the failure to see this here may just reflect differences between the development of the aneuploidy and the timing used here.

Response: This is an excellent point and we have updated the Discussion to discuss this important caveat to our conclusions.

Re: Spectrum00536-25R1 (**Loss of the *Aspergillus fumigatus* Spindle Assembly Checkpoint components, SldA or SldB, generates triazole heteroresistant conidial populations**)

Dear Dr. Jarrod R. Fortwendel:

Your manuscript has been accepted, and I am forwarding it to the ASM production staff for publication. Your paper will first be checked to make sure all elements meet the technical requirements. ASM staff will contact you if anything needs to be revised before copyediting and production can begin. Otherwise, you will be notified when your proofs are ready to be viewed.

Sincerely,
James Konopka
Editor
Microbiology Spectrum